# Structure–Activity Relationship of Aloperine Derivatives as New Anti–Liver Fibrogenic Agents

**DOI:** 10.3390/molecules25214977

**Published:** 2020-10-27

**Authors:** Kun Wang, Zhihao Guo, Yunyang Bao, Yudong Pang, Yinghong Li, Hongwei He, Danqing Song

**Affiliations:** Institute of Medicinal Biotechnology, Chinese Academy of Medical Sciences and Peking Union Medical College, Beijing 100050, China; kunwang224@163.com (K.W.); guozhihao96@163.com (Z.G.); 17714333891@163.com (Y.B.); pyd101101@163.com (Y.P.); songdanqing@imb.pumc.edu.cn (D.S.)

**Keywords:** aloperine, structure−activity relationship, COL1A1, anti-liver fibrogenesis

## Abstract

Twenty-seven novel 12*N*-substituted aloperine derivatives were synthesized and investigated for their inhibitory effects on collagen α1 (I) (COL1A1) promotor in human hepatic stellate LX-2 cells, taking aloperine (**1**) as the hit. A structure-activity relationship (SAR) study disclosed that the introduction of suitable substituents on the 12*N* atom might enhance the activity. Compound **4p** exhibited a good promise on down-regulating COL1A1 expression with the IC_50_ value of 16.5 μM. Its inhibitory activity against COL1A1 was further confirmed on both mRNA and protein levels. Meanwhile, it effectively inhibited the expression of other fibrogenic proteins, such as transforming growth factor β1 (TGF-β1) and smooth muscle actin (α-SMA). It also exhibited good in vivo safety profile with the oral LD_50_ value of 400 mg kg^−1^ in mice. The results initiated the anti-liver fibrogenic study of aloperine derivatives, and the key compound **4p** was selected as a novel lead for further investigation against liver fibrogenesis.

## 1. Introduction

Liver fibrosis, as a key pathogenic factor in chronic liver disease progression, is a consequence of chronic damage associated with alcoholic or non-alcoholic fatty liver disease, hepatitis, metabolic or genetic diseases [1,2]. Hepatic fibrosis is characterized by the dysregulated deposition of extracellular matrix (ECM), complexed with progressive destruction of normal liver tissue [2]. Without any treatment, hepatic fibrosis could progress to liver cirrhosis, hepatocellular carcinoma, and even hepatic failure [3]. However, up to now, no effective therapy is available for liver fibrosis except the elimination of underlying etiology or liver transplantation [4,5]. Therefore, the discovery of an effective antifibrotic therapy is still an unmet clinical need.

Transforming growth factor β1 (TGF-β1), the major fibrogenic cytokine, plays critical roles in the activation of hepatic stellate cells (HSCs) into myofibroblasts [6,7,8,9,10]. Active myofibroblast-like cells are characterized by increased migration, α-smooth muscle actin (α-SMA) expression, and robust collagen production, wherein type I collagen (COL1) constitutes the main source of extracellular matrix (ECM) in clinical and experimental liver fibrosis [11,12]. Taking COL1A1 promotor as the biomarker, a luciferase screening cell model based on the elevation effect of TGF-β1 upon the expression of COL1A1 promoter was established earlier in our group [13], and successfully applied to the screening and evaluation of anti-hepatic fibrosis drug candidates [14,15,16]. The in vivo pharmacodynamics study confirmed that compounds down-regulating the transcription of COL1A1 gene could effectively reverse liver fibrosis in vivo [15].

Aloperine (**1**, Figure 1), an endocyclic quinolizidin alkaloid isolated from the species *Sophora alopecuroides*, has a wide range of biological activities, such as anti-pulmonary fibrosis, anti-cancer, anti-inflammation, anti-virus and so on [17,18,19,20,21,22]. In view of its unique endocyclic and flexible structural scaffold, as well as a moderate potency in suppressing pulmonary fibrosis, compound **1** was taken as the hit to explore its effect on liver fibrosis. However, the anti-COL1A1 evaluation results indicated that it only held a mild activity by giving inhibitory rate of 4.7% at the concentration of 40 μM. Therefore, structural modification and optimization on **1** was carried out in this paper in the aim of achieving effective anti-liver fibrogenic leads.

In the present study, the anti-COL1A1 structure-activity relationship (SAR) of aloperine derivatives were explored for the first time, and twenty-seven novel aloperine derivatives with structural diversities on the 12*N* substituents were designed, synthesized and evaluated for their anti-COL1A1 activities.

## 2. Results and Discussion

### 2.1. Chemistry

The synthetic routes of all target compounds are depicted in Scheme 1. The target 12*N*-carbamoylmethyl aloperine derivatives **4a**–**p** were obtained by alkylation of **1** with the key intermediates **3a**–**p**, which were synthesized via acylation of bromoacetyl bromide with substituted amines **2a**–**p** under alkaline conditions [15].

Similarly, the acylation of substituted sulfenamides **5a**–**d** with bromoacetyl bromide in refluxing toluene afforded the key intermediates **6a**–**d**, which were alkylated with **1** to achieve the target 12*N*-sulfonylcarbamoylmethyl aloperine derivatives **7a**–**d** in total yields of 12–40%. At the meantime, glycine was sulfonylated by different sulfonyl chlorides **8a**–**g** to produce intermediates **9a**–**g**, and the coupling of **9a**–**g** with **1** in the presence of EDCI/HOBt formed the target 12*N*-sulfonylaminoacetyl aloperine compounds **10a**–**g**, in total yields of 22–51%.

### 2.2. SAR for Inhibition of COL1A1 Promotor of the Target Compounds in Human LX-2 Cells

A single luciferase reporter gene detection model was applied to evaluate the inhibitory effects of all target compounds on the expression of COL1A1 promotor in human hepatic stellate LX-2 cells at the concentrations of 40 and 80 μM, taking EGCG as the positive control [23]. The LX-2 cell lines were transfected with COL1A1 promotor luciferase plasmid pGL4.17-COL1A1P for 24 h, then simultaneously treated with TGF-β1 and a tested compound for 24 h [13]. The structures and inhibitory effects (%) of all target compounds were shown in Table 1.

Initially, the carbamoylmethyl moiety was selected as the linker, and a series of new 12*N*-carbamoylmethyl aloperine derivatives **4a**–**p** and **7a**–**d** with various substitutions were generated and evaluated. The introduction of an alkyl motiety, as in the cases of ethyl (**4a**), tetrahydro-2*H*-pyran-4-yl (**4b**) and 1-adamantyl (**4c**) lead to a slight increase in activity as compared to **1**. Among them, **4c** with a bulky 1-adamantyl group gave the higher activity, indicating that the introduction of a bulky motif might be beneficial for activity. At the meantime, the alkyl moiety was changed into a substituted benzene moiety to provide compounds **4d**–**l**. Most of them showed increased activity as compared to compound **1**. Especially, compounds **4h** and **4i** completely inhibited the expression of COL1A1 promotor with inhibitory rates of 99.9% at concentrations of 40 and 80 μM. These results indicated that the introduction of a benzene moiety on the 12*N* atom might be beneficial for activity.

Then, the secondary amine on the carbamoylmethyl linker was transformed into a tertiary amine, and compounds **4m**–**p** were created. Compounds **4n** and **4p** with bulky substituents displayed higher activity, with inhibitory rates of 99.9% and 72.2% at a concentration of 40 μM, and inhibitory rates of 100.0% and 99.1% at a concentration of 80 μM, respectively. These results indicated again that the introduction of a bulky motif on the 12*N* atom might be helpful for activity.

Furthermore, to investigate the possible effect of carbamoylmethyl linker on activity, an additionally sulfonyl group was attached and compounds **7a**–**d** were generated. The results showed that compounds **7a–d** showed a slight increase in activity. As a comparison, compounds **7a** and **4d**, differing from each other in a sulfonyl group, gave comparable activities, indicating that the insertion of a sulfonyl between the benzene ring and the carbamoylmethyl linker had little effect on potency. At the meantime, an aminoacetyl linker was introduced instead to achieve compounds **10a**–**g**. The anti-COL1A1 activity results showed that these compounds only gave moderate increases in activity, except **10g**. Notably, compounds **10a** and **10c** gave comparable activity to their counterparts **4a** and **4d**, respectively, while compound **10f** showed decreased activity compared to its counterpart **4g**, indicating that carbamoylmethyl linker on the 12*N* atom might be beneficial.

In the next step, compounds with diverse structural fragments and higher potencies, namely, **4h**, **4i**, **4n**, **4p** and **10g** were selected as representative compounds to examine their acute effects on luciferase activity, and the result showed that they had minimal inhibition on luciferase at the concentration of 80 μM after a 2-h treatment (Appendix A), indicating their direct effects on COL1A1 promotor. Then, their anti-COL1A1 potencies were verified by testing the half maximal inhibition concentration (IC_50_) values upon COL1A1 expression expression in LX-2 cells. As displayed in Table 1 and Appendix A, compounds **4i**, **4n** and **4p** gave the IC_50_ values of 8.4, 13.3 and 16.5 μM, respectively, while compound **4h** failed to give a dose-dependent inhibition within the given concentration range, which was probably due to its high cytotoxicity as shown in Table 2 and Appendix A. Compounds **4i**, **4n**, **4p** and **10g** gave higher median cytotoxic concentration (CC_50_) values of 19.5, 17.1, 34.8 and 68.6 μM in LX-2 cells, and 22.0, 35.2, 43.7 and 113.4 μM in HepG2 cells, respectively. Then, compounds **4i**, **4n** and **4p** with the lower IC_50_ values were selected as the key compounds for further studies.

### 2.3. Key Compounds Inhibited the Expression of COL1A1 on mRNA and Protein Levels

In the absence or presence of TGF-β1 (2 ng/mL) treatment, LX-2 cells were treated with key compounds **4n** and **4p** at concentrations of 6 μM and 12 μM, or **4i** at a concentration of 12 μM, respectively.

As shown in Figure 2A and Appendix A, in the absence of TGF-β1, real-time PCR (RT-PCR) amplification results indicated that the key compounds might lower the transcription of *COL1A1*, and compound **4n** gave inhibitory rates of 42% and 47% at concentrations of 6 μM and 12 μM, compound **4p** gave inhibitory rates of 38% and 42% at concentrations of 6 μM and 12 μM, and compound **4i** gave an inhibitory rate of 45%, respectively. At the meantime, upon the stimulation of TGF-β1, the transcription of *COL1A1* was significantly upregulated up to three times higher than that of the control (*p* < 0.01). As expected, the administration of the three tested compounds repressed the *COL1A1* transcription very significantly (*p* < 0.01) and compound **4n** gave inhibitory rates of 76.0% and 84.5% at concentrations of 6 μM and 12 μM, compound **4p** gave inhibitory rates of 99.5% and 102.9% at concentrations of 6 μM and 12 μM, and compound **4i** gave an inhibitory rate of 97.4%, respectively.

Then the anti-COL1A1 effects of **4n** and **4p** on protein level were evaluated by western blot assay. Without TGF-β1, there was only tiny amounts of COL1A1 expressed in LX2 cells, while the TGF-β1 treatment upregulated the expression of COL1A1 significantly. As displayed in Figure 3A–C and Appendix A, compound **4n** dose-dependently reversed the increase of COL1A1 protein with the inhibition rates of 31.8% and 76.5%, and compound **4p** gave the dose-dependent inhibition rates of 22.7% and 79.2% at the concentrations of 6 μM and 12 μM, respectively. These results suggested that these key compounds could effectively reduce COL1A1 expression on both mRNA and protein levels.

### 2.4. Key Compounds Inhibited the Expression of TGF-β on mRNA and Protein Levels

Since the stimulation of TGF-β1 provokes the expression of a series of fibrosis genes, for example *COL1A1* and *TGFB1* [24,25], the effects of key compounds on TGF-β1 expression were evaluated by RT-PCR and western blot assays. As shown in Figure 2B and Appendix A, the expression of *TGFB1* mRNA was maintained at a low level, and the addition of a key compound seemed to have little effect on its expression. Then, the addition of TGF-β1 caused an abnormally increased expression of *TGFB1* itself on mRNA level, which could be reversed significantly by the treatments of **4n**, **4p** or **4i.** Compound **4n** gave inhibitory rates of 57.7% and 60.0% at concentrations of 6 μM and 12 μM, compound **4p** gave inhibitory rates of 60.8% and 63.8% at concentrations of 6 μM and 12 μM, and compound **4i** gave an inhibitory rate of 75.4%, respectively, which showed a good consistency with the above anti-COL1A1 results.

At the meantime, the addition of TGF-β1 also stimulated the expression of TGF-β1 on protein level, as shown in Figure 3A–C and Appendix A. Both **4n** and **4p** treatment displayed a dose-dependent inhibition on TGF-β1 expression, compound **4n** gave inhibition rates of 14.6% and 61.0%, and compound **4p** gave dose-dependent inhibition rates of 20.8% and 135.4% at concentrations of 6 μM and 12 μM, respectively.Therefore, **4n** and **4p** could inhibit TGF-β1 expression and might block fibrogenesis due to the essential role of TGF-β1 in fibrogenesis [6].

### 2.5. Key Compounds Inhibited the Expressions of α-SMA on Protein Level

Stellate cell activation is a key process for liver fibrogenesis, and α-SMA expression, induced by a stimulation of TGF-β1 or other factors, is considered a reliable marker of hepatic stellate cell activation and a key biomarker for liver fibrosis [7,26]. Therefore, the expression of α-SMA was also investigated. As presented in Figure 3A–C and Appendix A, in our experiments, the addition of TGF-β1 stimulated the expression of α-SMA on protein level as expected. Similarly, a dose-dependent inhibition of α-SMA expression were observed in both treatments of **4n** and **4p** These results disclosed that the key compounds might also inhibited the expression of fibrogenic α-SMA.

### 2.6. Safety Profile of Key Compounds

To evaluate the safety profiles of the key compounds **4i**, **4n** and **4p**, an acute toxicity test was performed in Kunming mice. Compound **4i** was given orally in a single-dosing experiment at 100, 300 and 400 mg kg^−1^, while **4n** at 300, 400, 500 mg kg^−1^, and **4p** at 200, 400, 500 mg kg^−1^. The mice were closely monitored for 7 days. Compound **4n** and **4p** gave their median lethal dose (LD_50_) values of 500 mg kg^−1^ and 400 mg kg^−1^, as shown in Table 3, indicating good safety profiles in vivo.

### 2.7. Discussion

There are few studies on the anti-fibrotic effect of compound **1,** except one by Yin et al. that found that aloperine showed anti-pulmonary fibrosis by attenuating fibroblast proliferation and piferentiation via repressed PI3K/AKT/mTOR signaling and TGF-β/Smad signaling [22]. In view of the similar pathogenesis between fibrosis among tissues, our current study elucidated the anti-liver fibrogenesis effect of aloperine and its kind, in the aim of exploring the potential role of aloperine analogues in other types of tissues fibrosis.

Overexpression of collagen is a basic feature of liver fibrosis, and also an important cause of organic lesions [7,11,12]. Taking advantage of this mechanism, Stefanovic et al. [27] established a high-throughput screening model of antifibrotic drugs targeting protein-protein interaction between type I collagen 5′SL and larp6, based on the fact that the COL1A1 5′SL and Arp6 interaction plays an important role in collagen production. In our design to construct a COL1A1-based model, the high expression of COL1A1 in various liver fibrosis processes especially at the transcriptional level was considered. Therefore, the selected compounds by this model inhibited the activity of COL1A1 promotor and thus might hold the advantage of dealing with multi-type tissue fibrosis.

Through systematic structural modification and optimization, aloperine derivatives **4p** was identified to show high potency on inhibiting the COL1A1 promoter and suppressing the expression of fibrogenic proteins COL1A1, TGF-β1, and α-SMA, with low cytotoxicity and high LD_50_ values in acute toxicity test, indicating the promise against liver fibrogenesis. The down-regulating COL1A1 mechanism of aloperine derivatives deserves further investigation.

## 3. Materials and Methods

### 3.1. Apparatus, Materials, and Analysis Reagents

All solvents and chemical reagents were used as purchased without any further treatment. Aloperine was purchased from the Yanchi Dushun Biological and Chemical Co. Ltd. (Shanxi, China), with the purity over 95%. Room temperature (rt) refers to 20–25 °C. Melting point (mp) was determined on MP90 melting point apparatus and was uncorrected (Mettler-Toledo, Greifensee, Switzerland). ^1^H-NMR and ^13^C-NMR spectra were recorded on a Avance 400 MHz or 500 MHz (Bruker, Zürich, Switzerland) and 600 MHz instruments (Bruker, Zürich, Switzerland), taking DMSO-*d_6_* or CDCl_3_, as the solvent. ESI high-resolution mass spectrometry (ESI-HRMS) was performed on an AutospecUitima-TOF mass spectrometer (Micromass UK Ltd., Manchester, UK). Flash chromatography was performed on a Combiflash Rf 200 system (Teledyne, Lincoln, NE, USA).

### 3.2. Chemistry

#### 3.2.1. General Procedure for 12N-Substituted Carbamoylmethyl Aloperine Derivatives (**4a**–**p**)

To a solution of substituted amine **2a**–**p** (2 mmol) in anhydrous dichloromethane (30 mL), triethylamine (0.3 mL, 2 mmol) was added and stirred at 0 °C, and then bromoacetyl bromide (0.4 g, 2 mmol) was slowly added. After 30 min, the reaction system was warmed to room temperature, stirred for 1.5 h, then triethylamine (0.3 mL, 2 mmol) and **1** (0.5 g, 2 mmol) were added and stirred overnight at room temperature until the TLC analysis showed completion of the reaction. The resulting mixture was washed with water and brine, dried over anhydrous Na_2_SO_4_, filtered and concentrated to give a residue, which was purified by flash column chromatography on silica gel with dichloromethane/methanol as the eluent to furnish the title compounds.

*12N-N′-Ethylcarbamoylmethyl aloperine* (**4a**), Yield 38%; brown solid; m.p.: 69–71 °C. ^1^H-NMR (400 MHz, CDCl_3_) δ 9.16 (s, 1H), 9.01 (s, 1H), 5.85 (d, *J* = 5.2 Hz, 1H), 4.60–4.40 (m, 2H), 4.32–4.16 (m, 1H), 4.15–4.04 (m, 1H), 3.99–3.72 (m, 3H), 3.61–3.44 (m, 2H), 3.42–3.32 (m, 1H), 3.29–3.16 (m, 3H), 2.98–2.81 (m, 1H), 2.60–2.49 (m, 1H), 2.43–2.34 (m, 1H), 2.32–2.21 (m, 2H), 2.18–2.06 (m, 1H), 2.01–1.85 (m, 2H), 1.83–1.64 (m, 4H), 1.56–1.42 (m, 1H), 1.20 (s, 1H), 1.14 (t, *J* = 7.2 Hz, 3H). ^13^C-NMR (101 MHz, CDCl_3_) δ 163.4, 135.7, 127.7, 63.5, 59.3, 56.0, 55.1, 53.3, 46.2, 34.7, 33.6, 30.7, 27.7, 24.2, 23.5, 22.6, 21.2, 18.4, 14.3. ESI-HR-MS: calcd. for: C_19_H_32_N_3_O [M + H]^+^, 318.2540, found: 318.2530.

*12N-N′-(Tetrahydro-2H-pyran-4-yl)carbamoylmethyl aloperine hydrochloride* (**4b**), Yield 29%; white solid; m.p.: 199–201 °C. ^1^H-NMR (400 MHz, CDCl_3_) δ 11.16 (s, 1H), 9.52 (d, *J* = 5.2 Hz, 1H), 9.16 (s, 1H), 5.88 (d, *J* = 5.2 Hz, 1H), 4.71–4.58 (m, 1H), 4.44 (s, 1H), 4.11–4.04 (m, 1H), 4.01–3.91 (m, 4H), 3.90–3.83 (m, 2H), 3.76 (s, 1H), 3.68–3.53 (m, 2H), 3.48–3.38 (m, 2H), 3.36–3.27 (m, 2H), 3.22–3.10 (m, 1H), 3.09–2.95 (m, 1H), 2.63–2.52 (m, 1H), 2.41 (s, 1H), 2.36–2.21 (m, 2H), 2.19–2.06 (m, 1H), 2.04–1.97 (m, 1H), 1.91–1.80 (m, 3H), 1.80–1.68 (m, 4H), 1.56–1.43 (m, 1H). ^13^C-NMR (101 MHz, CDCl_3_) δ 162.9, 135.9, 127.6, 66.5 (2), 63.6, 59.5, 56.4, 55.2, 53.4 46.5, 46.4, 33.6, 32.2, 32.2, 30.8, 27.8, 24.3, 23.6, 22.6, 21.1, 18.5. ESI-HR-MS: calcd. for: C_22_H_36_N_3_O_2_ [M + HCl–Cl]^+^, 374.2802, found: 374.2795.

*12N-N′-Adamantylcarbamoylmethyl aloperine* (**4c**), Yield 17%; brown solid; m.p.: 176–178 °C. ^1^H-NMR (400 MHz, CDCl_3_) δ 7.31 (s, 1H), 5.67 (s, 1H), 3.87–3.73 (m, 1H), 3.44–3.32 (m, 1H), 3.26–3.12 (m, 2H), 3.11–2.95 (m, 3H), 2.95–2.82 (m, 2H), 2.59–2.43 (m, 3H), 2.31–2.15 (m, 3H), 2.10–2.03 (m, 8H), 1.96–1.89 (m, 5H), 1.70–1.61 (m, 9H), 1.56–1.48 (m, 2H). ^13^C-NMR (101 MHz, CDCl_3_) δ 169.4, 140.3, 122.3, 63.1, 61.8, 58.1, 57.1, 56.0, 54.2, 52.2, 51.2, 45.5, 41.9 (3), 36.4 (3), 33.8, 33.1, 29.5 (3), 25.7, 23.8, 22.8, 18.8. ESI-HR-MS: calcd. for: C_27_H_42_N_3_O [M + H]^+^, 424.3322, found: 424.3311.

*12N-N′-Phenylcarbamoylmethyl aloperine* (**4d**), Yield 55%; white solid; m.p.: 66–68 °C. ^1^H-NMR (400 MHz, CDCl_3_) δ 7.19 (t, *J* = 7.6 Hz, 2H), 6.71 (t, *J* = 7.2 Hz, 1H), 6.63 (d, *J* = 8.0 Hz, 2H), 5.64 (d, *J* = 4.0 Hz, 1H), 4.95 (s, 1H), 4.75 (d, *J* = 4.0 Hz, 1H), 4.01–3.93 (m, 1H), 3.88–3.80 (m, 1H), 3.79–3.67 (m, 1H), 3.47–3.39 (m, 1H), 3.09–3.01 (m, 1H), 2.74 (d, *J* = 8.0 Hz, 2H), 2.63–2.55 (m, 1H), 2.44 (s, 1H), 2.39–2.31 (m, 1H), 2.31–2.21 (m, 1H), 2.17–2.06 (m, 1H), 1.97 (s, 1H), 1.94–1.83 (m, 4H), 1.83–1.76 (m, 1H), 1.69 (m, 2H), 1.51–1.39 (m, 1H), 1.15–1.02 (m, 2H). ^13^C-NMR (101 MHz, CDCl_3_) δ 167.8, 147.6, 135.4, 129.4 (2), 128.4, 117.5, 113.1 (2), 59.4, 59.2, 54.4, 46.9 45.5, 40.6, 35.1, 31.5, 28.0, 26.3 24.9, 24.6, 24.1, 19.2. ESI-HR-MS: calcd. for: C_23_H_32_N_3_O [M + H]^+^: 366.2540, found: 366.2541.

*12N-N′-(4-Fluorophenyl)carbamoylmethyl aloperine* (**4e**), Yield 50%; white solid; mp 119–121 °C. ^1^H-NMR (500 MHz, CDCl_3_) δ 9.25 (s, 1H), 7.57–7.43 (m, 2H), 7.01 (t, *J* = 8.5 Hz, 2H), 5.62 (d, *J* = 5.0 Hz, 1H), 3.68–3.52 (m, 1H), 2.92 (d, *J* = 5.0 Hz, 2H), 2.80–2.71 (m, 1H), 2.70–2.62 (m, 2H), 2.60–2.52 (m, 1H), 2.48–2.42 (m, 1H), 2.34–2.24 (m, 2H), 2.22–2.14 (m, 1H), 2.08–1.97 (m, 2H), 1.93–1.84 (m, 2H), 1.80–1.66 (m, 2H), 1.64–1.50 (m, 2H), 1.50–1.42 (m, 2H), 1.42–1.35 (m, 1H), 1.34–1.23 (m, 2H); ^13^C-NMR (126 MHz, CDCl_3_) δ 169.5, 159.4, 134.1, 133.0, 129.3, 121.0, 121.0, 116.0, 115.8, 64.3, 63.9, 57.3, 55.9, 55.3, 51.7, 35.8, 33.4, 31.6, 30.3, 26.2, 25.5, 24.8, 23.8. ESI-HR-MS: calcd. for: C_23_H_31_FN_3_O [M + H]^+^: 384.2446, found: 384.2443.

*12N-N′-(3-Methylphenyl)carbamoylmethyl aloperine* (**4f**), Yield 51%; white solid; mp 40–42 °C. ^1^H-NMR (500 MHz, CDCl_3_) δ 9.19 (s, 1H), 7.39 (s, 1H), 7.32 (d, *J* = 7.5 Hz, 1H), 7.21 (t, *J* = 8.0 Hz, 1H), 6.91 (d, *J* = 7.0 Hz, 1H), 5.64 (s, 1H), 3.65–3.56 (m, 1H), 3.09 (s, 1H), 2.92 (s, 2H), 2.79–2.65 (m, 2H), 2.59 (s, 1H), 2.52–2.45 (m, 1H), 2.34 (s, 3H), 2.31–2.15 (m, 2H), 2.09–1.99 (m, 2H), 1.92 (s, 2H), 1.81–1.73 (m, 1H), 1.72–1.66 (m, 1H), 1.65–1.54 (m, 2H), 1.52–1.44 (m, 2H), 1.44–1.36 (m, 1H), 1.36–1.22 (m, 2H), 0.90–0.81 (m, 1H); ^13^C-NMR (126 MHz, CDCl_3_) δ 169.2, 139.0, 137.7, 132.9, 129.0, 128.9, 124.9, 119.8, 116.3, 63.6, 57.2, 55.6, 55.0, 51.4, 35.5, 33.2, 31.4, 30.0, 26.9, 25.9, 25.2, 24.6, 23.5, 21.5. ESI-HR-MS: calcd. for: C_24_H_34_N_3_O [M + H]^+^: 380.26964, found: 380.2687.

*12N-N′-(m-Trifluoromethoxyphenyl)carbamoylmethyl aloperine* (**4g**), Yield 40%; light brown solid; mp 50–52 °C. ^1^H-NMR (500 MHz, CDCl_3_) δ 10.16 (s, 1H), 8.30 (s, 1H), 8.11 (d, *J* = 7.5 Hz, 1H), 7.31 (m, 1H), 6.93 (d, *J* = 7.5 Hz, 1H), 5.68 (d, *J* = 5.5 Hz, 1H), 3.65–3.52 (m, 1H), 3.52–3.42 (m, 1H), 3.38–3.30 (m, 1H), 3.30–3.22 (m, 1H), 3.20–3.05 (m, 5H), 2.86–2.75 (m, 1H), 2.61–2.49 (m, 2H), 2.38–2.27 (m, 2H), 2.13–1.89 (m, 4H), 1.80–1.59 (m, 4H), 1.58–1.41 (m, 2H); ^13^C-NMR (126 MHz, CDCl_3_) δ 169.5, 149.2, 140.9, 139.9, 129.9, 121.8, 119.3, 116.1, 113.1, 61.7, 58.7, 56.8, 56.5, 56.4, 53.5, 45.5, 33.8, 33.7, 29.4, 24.2, 23.4, 23.0, 22.6, 18.7. ESI-HR-MS: calcd. for: C_24_H_31_F_3_N_3_O_2_ [M + H]^+^: 450.2363, found: 450.2351.

*12N-N′-[4-(1H-Indol-2-yl)phenyl]carbamoylmethyl aloperine* (**4h**), Yield 18%; brown solid; m.p.: 219–221 °C. ^1^H-NMR (500 MHz, CDCl_3_) δ 9.77 (s, 1H), 9.32 (s, 1H), 8.20 (d, *J* = 7.5 Hz, 2H), 7.72–7.57 (m, 3H), 7.53–7.46 (m, 1H), 7.19–7.14 (m, 1H), 7.13–7.07 (m, 1H), 6.73 (s, 1H), 5.62 (s, 1H), 3.60 (s, 1H), 3.45–3.31 (m, 1H), 3.27–3.06 (m, 3H), 3.02–2.85 (m, 4H), 2.83–2.72 (m, 1H), 2.58–2.40 (m, 1H), 2.36–2.20 (m, 3H), 2.03–1.77 (m, 4H), 1.68–1.48 (m, 5H), 1.40–1.24 (m, 1H). ^13^C-NMR (126 MHz, CDCl_3_) δ 169.0, 140.5, 138.0, 137.4, 136.8, 129.1, 128.0, 125.6 (2), 122.0, 121.7, 120.9, 120.1, 119.9, 111.1, 98.7, 61.6, 58.2, 56.7, 56.5, 56.3, 53.3, 45.3, 33.6, 33.3, 29.1, 23.8, 23.3, 22.6, 22.1, 18.4. ESI-HR-MS: calcd for C_31_H_37_N_4_O [M + H]^+^, 481.2962, Found: 481.2964.

*12N-N′-(3-Chloro-5-trifluoromethoxyphenyl)carbamoylmethyl aloperine* (**4i**), Yield 35%; light pink solid; m.p.: 44–46 °C. ^1^H-NMR (400 MHz, CDCl_3_) δ 9.53 (s, 1H), 7.55 (s, 1H), 7.47 (s, 1H), 6.96 (s, 1H), 5.63 (d, *J* = 5.2 Hz, 1H), 3.63–3.55 (m, 1H), 2.97–2.84 (m, 2H), 2.81–2.73 (m, 1H), 2.72–2.60 (m, 2H), 2.54–2.46 (m, 1H), 2.46–2.38 (m, 1H), 2.36–2.21 (m, 3H), 2.10–1.95 (m, 3H), 1.93 (s, 1H), 1.81–1.74 (m, 1H), 1.72–1.60 (m, 2H), 1.59–1.49 (m, 2H), 1.45–1.28 (m, 4H). ^13^C-NMR (101 MHz, CDCl_3_) δ 170.0, 149.9, 140.0, 135.5, 133.3, 129.0, 120.4, 117.5, 116.6, 110.2, 63.7, 63.3, 57.2, 55.7, 55.4, 51.0, 35.6, 33.3, 30.6, 30.4, 25.9, 25.4, 24.0, 23.9. ESI-HR-MS: calcd. for: C_24_H_30_ClF_3_N_3_O_2_ [M + H]^+^, 484.1973, found: 484.1965.

*12N-N′-(2-Methyl-5-methoxycarbonylphenyl)carbamoylmethyl aloperine* (**4j**), Yield 16%; violet solid; m.p.: 75–77 °C. ^1^H-NMR (500 MHz, CDCl_3_) δ 9.46 (s, 1H), 8.51–8.41 (m, 1H), 7.99–7.83 (m, 2H), 5.72–5.61 (m, 1H), 3.90 (s, 3H), 3.77–3.67 (m, 1H), 3.03–2.90 (m, 2H), 2.84–2.72 (m, 1H), 2.65 (s, 2H), 2.49 (s, 1H), 2.39–2.13 (m, 6H), 2.11–1.99 (m, 3H), 1.92 (s, 1H), 1.86–1.67 (m, 3H), 1.61–1.38 (m, 5H), 1.36–1.22 (m, 2H). ^13^C-NMR (126 MHz, CDCl_3_) δ 169.7, 167.0, 140.6, 132.6, 131.9, 129.6, 129.1, 125.5, 125.2, 119.2, 64.9, 63.8, 57.7, 56.0, 55.2, 52.2, 52.1, 35.7, 33.4, 32.0, 30.1, 26.1, 25.4, 25.0, 23.7, 18.2. ESI-HR-MS: calcd for C_26_H_36_N_3_O_3_ [M + H]^+^, 438.2751, Found: 438.2752.

*12N-N′-(3,4,5-Trimethylphenyl)carbamoylmethyl aloperine* (**4k**), Yield 42%; white solid; m.p.: 183–185 °C. ^1^H-NMR (400 MHz, CDCl_3_) δ 7.70 (s, 2H), 5.74 (s, 1H), 3.69–3.48 (m, 3H), 3.38–3.30 (m, 3H), 3.15–3.02 (m, 2H), 2.60–2.52 (m, 1H), 2.32 (s, 1H), 2.26 (s, 8H), 2.12–2.04 (m, 6H), 1.98–1.88 (m, 2H), 1.75–1.66 (m, 4H), 1.65–1.57 (m, 2H), 1.54–1.44 (m, 1H), 1.24 (s, 1H). ^13^C-NMR (101 MHz, CDCl_3_) δ 167.5, 139.7, 137.2, 136.9, 135.3, 131.2, 123.2, 119.5, 119.2, 62.1, 58.6, 56.4, 56.2, 56.0, 45.7, 33.8, 33.1, 29.1, 24.1, 23.1, 22.6, 20.8 (3), 18.6, 15.0. ESI-HR-MS: calcd. for: C_26_H_38_N_3_O [M + H]^+^, 408.3009, found: 408.2999.

*12N-N′-(5-Quinolyl)carbamoylmethyl aloperine* (**4l**), Yield 29%; yellow solid; m.p.: 74–76 °C. ^1^H-NMR (400 MHz, CDCl_3_) δ 9.94 (s, 1H), 8.95 (d, *J* = 3.2 Hz, 1H), 8.34 (d, *J* = 7.2 Hz, 1H), 8.22 (d, *J* = 8.8 Hz, 1H), 7.92 (d, *J* = 8.4 Hz, 1H), 7.73 (t, *J* = 8.0 Hz, 1H), 7.46 (dd, *J* = 8.4, 4.0 Hz, 1H), 5.69 (d, *J* = 4.8 Hz, 1H), 3.86–3.75 (m, 1H), 3.12–3.04 (m, 1H), 3.00 (s, 1H), 2.93–2.84 (m, 1H), 2.83–2.74 (m, 1H), 2.72–2.52 (m, 3H), 2.43–2.29 (m, 2H), 2.17–2.01 (m, 3H), 1.99–1.86 (m, 2H), 1.82–1.71 (m, 2H), 1.70–1.60 (m, 1H), 1.57–1.45 (m, 2H), 1.44–1.22 (m, 4H). ^13^C-NMR (101 MHz, CDCl_3_) δ 169.7, 150.4, 148.8, 132.9, 132.7, 129.9, 129.4, 128.9, 126.2, 121.0, 120.9, 118.3, 64.3, 63.8, 57.6, 56.0, 55.3, 51.9, 35.7, 33.4, 31.3, 30.3, 26.2, 25.4, 24.5, 23.8. ESI-HR-MS: calcd. for: C_26_H_33_N_4_O [M + H]^+^, 417.2649, found: 417.2638.

*12N-[(N’-Methyl-N’-phenyl)carbamoylmethyl aloperine* (**4m**), Yield 55%; white solid; m.p.: 73–75 °C. ^1^H-NMR (400 MHz, CDCl_3_) δ 7.22 (t, *J* = 7.6 Hz, 2H), 6.77–6.67 (m, 3H), 5.61 (d, *J* = 4.8 Hz, 1H), 4.71 (d, *J* = 3.2 Hz, 1H), 4.14–4.08 (m, 1H), 3.69 (s, 1H), 3.55–3.43 (m, 1H), 3.10–2.98 (m, 4H), 2.81–2.69 (m, 2H), 2.62–2.52 (m, 1H), 2.44–2.33 (m, 2H), 2.28–2.19 (m, 1H), 2.16–2.04 (m, 1H), 1.98–1.78 (m, 5H), 1.75–1.59 (m, 4H), 1.46–1.38 (m, 1H), 1.15–1.01 (m, 2H). ^13^C-NMR (101 MHz, CDCl_3_) δ 168.8, 149.6, 135.8, 129.3 (2), 128.1, 117.3, 112.7, 112.4, 59.5, 59.1, 55.0, 54.4, 46.9, 41.2, 39.7, 35.1, 31.4, 28.1, 26.3, 25.0, 24.9, 24.2, 19.3. ESI-HR-MS: calcd. for: C_24_H_34_N_3_O [M + H]^+^: 380.2696, found: 380.2696.

*12N-(N’-(Naphthalen-2-yl)-N’-phenyl)carbamoylmethyl aloperine* (**4n**), Yield 46%; white solid; m.p.: 79–81 °C. ^1^H-NMR (400 MHz, CDCl_3_) δ 7.89–7.74 (m, 3H), 7.69 (s, 1H), 7.48 (s, 2H), 7.42–7.34 (m, 3H), 7.33–7.28 (m, 3H), 5.48 (d, *J* = 5.6 Hz, 1H), 3.51–3.41 (m, 1H), 3.41–3.25 (m, 2H), 2.90 (d, *J* = 4.4 Hz, 2H), 2.54 (s, 1H), 2.32–2.15 (m, 3H), 2.07–1.92 (m, 3H), 1.82 (s, 1H), 1.76–1.67 (m, 2H), 1.66–1.59 (m, 2H), 1.55–1.46 (m, 3H), 1.41 (s, 1H), 1.30–1.20 (m, 3H). ^13^C-NMR (151 MHz, DMSO-*d_6_*) δ 163.7, 141.7, 139.3, 137.7, 134.4, 133.2, 130.3, 129.0, 128.6, 128.2, 127.7, 127.6, 127.4, 127.2, 127.0, 126.8, 126.0, 125.2, 124.9, 63.4, 58.1, 55.0, 53.8, 53.0, 44.6, 32.5, 30.1, 27.0, 22.9, 22.6, 21.9, 20.3, 17.7. ESI-HR-MS: calcd. for: C_33_H_38_N_3_O [M + H]^+^, 492.3009, found: 492.2994.

*12N-N′-(1-Pyrrolidinyl)carbamoylmethyl aloperine* (**4o**), Yield 50%; brown solid; m.p.: 78–80 °C. ^1^H-NMR (400 MHz, CDCl_3_) δ 9.61 (s, 1H), 6.02–5.78 (m, 1H), 5.74–5.48 (m, 2H), 5.07–4.85 (m, 1H), 4.67–4.54 (m, 1H), 4.48–4.32 (m, 1H), 3.99–3.84 (m, 1H), 3.79–3.71 (m, 1H), 3.69–3.59 (m, 2H), 3.56–3.37 (m, 4H), 3.37–3.25 (m, 2H), 3.21–2.84 (m, 2H), 2.62–2.48 (m, 1H), 2.40–2.28 (m, 2H), 2.27–2.12 (m, 1H), 2.09–1.88 (m, 4H), 1.87–1.79 (m, 3H), 1.78–1.56 (m, 3H), 1.54–1.40 (m, 1H). ^13^C-NMR (101 MHz, CDCl_3_) δ 161.9, 136.1, 127.4, 64.0, 59.5, 56.4, 55.2, 53.8, 46.5, 46.5, 46.0, 33.6, 30.8, 27.5, 26.1, 24.3, 24.1, 23.5, 22.7, 21.0, 18.4. ESI-HR-MS: calcd. for: C_21_H_34_N_3_O [M + H]^+^: 344.2696, found: 344.2683.

*12N-N′-Phenoxazinylcarbamoylmethyl aloperine* (**4p**), Yield 56%; light brown solid; m.p.: 84–86 °C. ^1^H-NMR (400 MHz, CDCl_3_) δ 7.50–7.44 (m, 2H), 7.22–7.16 (m, 2H), 7.15–7.08 (m, 4H), 5.48 (d, *J* = 5.2 Hz, 1H), 3.79–3.60 (m, 2H), 3.34 (s, 1H), 2.92–2.68 (m, 3H), 2.52–2.38 (m, 3H), 2.20–2.13 (m, 1H), 2.11–1.91 (m, 3H), 1.83 (s, 1H), 1.78–1.68 (m, 2H), 1.61–1.51 (m, 3H), 1.48–1.38 (m, 3H), 1.32–1.20 (m, 2H). ^13^C-NMR (101 MHz, CDCl_3_) δ 169.4, 151.3 (2), 133.5, 129.1 (2), 127.6 (2), 127.2, 125.2 (2), 123.6 (2), 117.3 (2), 66.1, 62.1, 55.9, 54.2, 52.9, 51.8, 35.6, 33.3, 32.9, 29.5, 26.3, 25.6, 25.1, 23.4. ESI-HR-MS: calcd. for: C_29_H_34_N_3_O_2_ [M + H]^+^, 456.2646, found: 456.2638.

#### 3.2.2. General Procedures for 12N-Substituted Sulfonylcarbamoylmethyl Aloperine Derivatives (**7a**–**d**)

To a solution of substituted sulfanilamide (2 mmol) in toluene (20 mL), bromoacetyl bromide (0.8 g, 4 mmol) was added dropwise, the mixture was stirred at 0 °C for 30 min, then reaction was stirred at 90 °C for 5 h. The reaction was monitored by TLC until TLC showed the completion of the reaction. Then the mixture was evaporated, and the gained residue was suspended in saturated NaHCO_3_ solution (40 mL). The mixture was filtered, and conc. HCl solution was added dropwise to the filtrate to precipitate the intermediates **6a***–***d**, which was applied to the next step without further purification. To a solution of **6a***–***d** (1 mmol) in DMF (30 mL), **1** (1.2 mmol) and K_2_CO_3_ (3 mmol) was added. The mixture was heated at 100 °C and stirred overnight. The mixture was evaporated, and the residue was dissolved in dichloromethane, filtered and purified by flash column chromatography on silica gel with dichloromethane/methanol as the eluent to give the title compounds **7a***–***d**.

*12N-N′-Benzenesulfonyl carbamoylmethyl aloperine* (**7a**), Yield: 40%; light yellow solid, m.p.: 113–115 °C. ^1^H-NMR (400 MHz, CDCl_3_) δ 7.99 (dd, *J* = 8.0, 1.6 Hz, 2H), 7.47–7.35 (m, 3H), 5.66 (d, *J* = 6.0 Hz, 1H), 3.55–3.45 (m, 1H), 3.36 (d, *J* = 4.8 Hz, 1H), 3.25–2.90 (m, 7H), 2.74–2.61 (m, 1H), 2.54 (s, 1H), 2.34–2.23 (m, 1H), 2.14–2.09 (m, 1H), 2.08–1.96 (m, 2H), 1.93–1.82 (m, 2H), 1.81–1.60 (m, 3H), 1.57–1.41 (m, 4H), 1.29–1.23 (m, 1H). ^13^C-NMR (101 MHz, CDCl_3_) δ 174.0, 144.3 137.6, 131.0, 128.2(2), 127.1(2), 125.8, 62.7, 59.5, 55.2, 54.6, 46.8, 34.1, 33.1, 31.7, 29.1, 24.9, 23.5, 22.7, 22.7, 19.6. ESI-HR-MS: calcd for C_23_H_32_N_3_O_3_S [M + H]^+^: 430.2159, found: 430.2147.

*12N-N′-3-Trifluoromethylbenzenesulfonyl carbamoylmethyl aloperine* (**7b**). Yield: 22%; light yellow solid, m.p.: 105–107 °C. ^1^H-NMR (400 MHz, CDCl_3_) δ 8.23–8.14 (m, 2H), 7.65 (d, *J* = 7.6 Hz, 1H), 7.52 (t, *J* = 7.6 Hz, 1H), 5.59 (d, *J* = 6.0 Hz, 1H), 3.45–3.36 (m, 1H), 3.34–3.14 (m, 7H), 2.97–2.88 (m, 1H), 2.56–2.40 (m, 2H), 2.30–2.22 (m, 1H), 2.16 (s, 1H), 2.05–1.84 (m, 5H), 1.83–1.75 (m, 1H), 1.66–1.46 (m, 5H). ^13^C NMR (101 MHz, CDCl_3_) δ 175.2, 146.0, 138.6, 130.7, 130.4, 128.7, 127.3, 125.1, 124.2, 123.9, 62.6(2), 60.1, 55.3, 54.5, 46.1, 34.0, 33.3, 29.2, 24.9, 24.1, 23.4, 23.2, 19.1. ESI-HR-MS: calcd for C_24_H_31_F_3_N_3_O_3_S [M + H]^+^: 498.2033, found: 498.2019.

*12N-N′-(4-Bromo-3-trifluoromethylbenzenesulfonyl)carbamoylmethyl aloperine (**7c**).* Yield: 12%; white solid, m.p.: 79–81 °C. ^1^H-NMR (600 MHz, CDCl_3_) δ 8.25 (s, 1H), 8.07 (d, *J* = 7.8 Hz, 1H), 7.74 (d, *J* = 7.8 Hz, 1H), 5.67 (d, *J* = 4.8 Hz, 1H), 3.49–3.16 (m, 6H), 3.15–3.06 (m, 1H), 3.02–2.90 (m, 1H), 2.61–2.39 (m, 2H), 2.37–2.27 (m, 1H), 2.21 (s, 1H), 2.07–1.83 (m, 5H), 1.64–1.47 (m, 5H), 1.30–1.19 (m, 3H). ^13^C-NMR (101 MHz, CDCl_3_) δ 173.4, 144.6, 140.2, 134.8, 132.0,129.8, 127.0, 123.7, 122.8, 122.6, 62.4, 60.2, 59.0, 55.2, 54.5, 46.1, 33.9, 33.4, 29.2, 24.4, 23.1, 23.0, 22.8, 18.7. ESI-HR-MS: calcd for C_254_H_30_BrF_3_N_3_O_3_S [M + H]^+^: 576.1138, found: 576.1122.

*12N-N′-(1-Naphtylsulfonyl)carbamoylmethyl aloperine* (**7d**). Yield: 31%; white solid, m.p.: 93–95 °C. ^1^H-NMR (600 MHz, CDCl_3_) δ 8.70 (s, 1H), 8.12 (dd, *J* = 8.4, 1.6 Hz, 1H), 8.02 (d, *J* = 7.8 Hz, 1H), 7.97 (d, *J* = 8.4 Hz, 1H), 7.90 (d, *J* = 8.4 Hz, 1H), 7.64 (t, *J* = 7.2 Hz, 1H), 7.60 (t, *J* = 7.2 Hz, 1H), 5.70 (d, *J* = 6.0 Hz, 1H), 3.89–3.77 (m, 1H), 3.52 (s, 1H), 3.48–3.36 (m, 3H), 3.36–3.26 (m, 3H), 3.00–2.92 (m, 1H), 2.75–2.63 (m, 2H), 2.42–2.35 (m, 1H), 2.31 (s, 1H), 2.13–2.00 (m, 4H), 2.00–1.91 (m, 2H), 1.75–1.65 (m, 3H), 1.65–1.60 (m, 1H), 1.58–1.47 (m, 2H). ^13^C-NMR (101 MHz, CDCl_3_) δ 173.3, 141.5, 138.7, 134.3, 132.4, 129.3, 128.1, 127.7, 127.7, 127.6, 126.7, 124.8, 123.8, 62.6 (2), 59.9, 55.3, 54.4, 46.4, 34.0, 33.2, 29.1, 24.6, 23.4, 23.3, 22.7, 19.1. ESI-HR-MS: calcd for C_27_H_34_N_3_O_3_S [M + H]^+^: 480.2315, found: 480.2311.

#### 3.2.3. General Procedures for 12N-sulfonylaminoacetylaloperine Derivatives **10a**–**g**

To a solution of glycine (3 mmol) and NaOH (3.6 mL) in H_2_O (15 mL), the substituted sulphonyl chloride **8a**–**g** (3.3 mmol) was slowly added at 0 °C, then the mixture was stirred at room temperature for 3 h. After the mixture was adjusted to pH 2 by 20% aqueous hydrochloric acid, brine (30 mL) was added, and the gained mixture was extracted with dichloromethane (30 mL × 2). The organic layer was dried over anhydrous Na_2_SO_4_ and concentrated under vacuum to afford intermediates **9a**–**g** without further purification.

To the solution of **9a**–**g** in dichloromethane (30 mL) at 0 °C, N-hydroxybenzotrizol (HOBt, 4 mmol), 1-ethyl-3-(3-dimethylaminopropyl) carbodiimide (EDCI, 6 mmol) and diisopropyl-ethylamine (1.3 mL) was added and stirred for 30 min, followed by addition of **1** (3 mmol). The reaction was warmed to room temperature, stirred for 12 h. The mixture was washed with water (50 mL × 2) and brine (50 mL), dried over anhydrous Na_2_SO_4_, filtered and then the filtrate was concentrated in vacuum. Then the residue was purified by flash column chromatography on silica gel with dichloromethane/methanol as the eluent to give target compounds **10a**–**g**.

*12N-N′-Ethylsulfonylaminoacetyl aloperine* (**10a**), Yield: 51%; white solid; mp 145–147 °C. ^1^H-NMR (400 MHz, CDCl_3_) δ 5.67–5.60 (m, 1H), 5.38–5.31 (m, 1H), 4.68 (d, *J* = 4.8 Hz, 1H), 4.07–3.99 (m, 1H), 3.93–3.84 (m, 1H), 3.77–3.65 (m, 1H), 3.31–3.21 (m, 1H), 3.13–2.99 (m, 3H), 2.73 (d, *J* = 7.4 Hz, 2H), 2.62–2.53 (m, 1H), 2.41–2.22 (m, 2H), 2.15–2.03 (m, 1H), 2.02–1.94 (m, 1H), 1.91–1.59 (m, 8H), 1.50–1.35 (m, 4H), 1.15–1.00 (m, 2H). ^13^C-NMR (101 MHz, CDCl_3_) δ 166.2, 134.9, 128.7, 59.4, 59.3, 54.4, 47.4, 46.7, 44.4, 40.8, 34.9, 31.4, 27.8, 26.1, 24.7, 24.5, 24.0, 19.2, 8.4. ESI-HR-MS: calcd for C_19_H_32_N_3_O_3_S [M + H]^+^: 382.2159, found: 382.2151.

*12N-N′-n-**Butylsulfonyl**aminoacetyl aloperine* (**10b**), Yield: 47%; white solid; m.p.: 98–100 °C. ^1^H-NMR (400 MHz, CDCl_3_) δ 9.90 (s, 1H), 5.80 (s, 1H), 4.76 (s, 1H), 4.22–3.98 (m, 2H), 3.94–3.78 (m, 2H), 3.63–3.26 (m, 3H), 3.21–2.96 (m, 4H), 2.84 (s, 1H), 2.75–2.57 (m, 1H), 2.33 (s, 1H), 2.29–2.12 (m, 2H), 2.10–2.02 (m, 1H), 2.01–1.89 (m, 2H), 1.86–1.66 (m, 7H), 1.61–1.51 (m, 1H), 1.50–1.39 (m, 2H), 1.27–1.21 (m, 1H), 0.94 (t, *J* = 7.2 Hz, 3H). ^13^C-NMR (101 MHz, CDCl_3_) δ 167.9, 137.5, 126.1, 58.2, 58.1, 54.8, 53.1, 45.6, 44.7, 43.0, 33. 9, 29.5, 27.2, 25.6, 24.0, 22.8, 22.5, 22.2, 21.6, 18.8, 13.7. ESI-HR-MS: calcd for C_21_H_36_N_3_O_3_S [M + H]^+^: 410.2472, found: 410.2461.

*12N-N′-**Benzenesulfonyl**aminoacetyl**aloperine* (**10c**), Yield: 45%; white solid; m.p.: 63–65 °C. ^1^H-NMR (400 MHz, CDCl_3_) δ 7.88 (d, *J* = 7.2 Hz, 2H), 7.58 (t, *J* = 7.2 Hz, 1H), 7.51 (t, *J* = 7.6 Hz, 2H), 5.83–5.74 (m, 1H), 5.59 (d, *J* = 4.8 Hz, 1H), 4.51 (d, *J* = 5.2 Hz, 1H), 3.86–3.78 (m, 1H), 3.75–3.68 (m, 1H), 3.65–3.55 (m, 1H), 3.19–3.11 (m, 1H), 2.95–2.87 (m, 1H), 2.70–2.63 (m, 2H), 2.59–2.48 (m, 1H), 2.29–2.12 (m, 2H), 2.09–2.02 (m, 1H), 1.99–1.90 (m, 2H), 1.88–1.75 (m, 3H), 1.71–1.64 (m, 2H), 1.63–1.57 (m, 2H), 1.48–1.34 (m, 1H), 1.14–0.97 (m, 2H).^13^C-NMR (101 MHz, CDCl_3_) δ 165.3, 139.2, 134.8, 132.9, 129.2 (2), 128.7, 127.5 (2), 59.4, 59.2, 54.3, 46.7, 43.8, 40.6, 35.0, 31.3, 27.8, 26.2, 24.7, 24.4, 23.9, 19.1.ESI-HR-MS: calcd for C_23_H_32_N_3_O_3_S [M + H]^+^: 430.2159, found: 430.2144.

*12N-N′-m-**Bromobenzenesulfonyl**aminoacetyl aloperine* (**10d**), Yield: 35%; white solid; m.p.: 101–103 °C. ^1^H-NMR (400 MHz, CDCl_3_) δ 8.02 (s, 1H), 7.82 (d, *J* = 7.6 Hz, 1H), 7.70 (d, *J* = 8.0 Hz, 1H), 7.39 (t, *J* = 8.0 Hz, 1H), 5.90–5.75 (m, 1H), 5.60 (d, *J* = 5.2 Hz, 1H), 4.53 (d, *J* = 5.6 Hz, 1H), 3.89–3.79 (m, 1H), 3.77–3.69 (m, 1H), 3.68–3.57 (m, 1H), 3.20–3.11 (m, 1H), 2.97–2.91 (m, 1H), 2.72–2.64 (m, 2H), 2.59–2.50 (m, 1H), 2.27–2.17 (m, 2H), 2.09–2.02 (m, 1H), 2.00–1.90 (m, 2H), 1.89–1.77 (m, 3H), 1.76–1.67 (m, 2H), 1.64–1.58 (m, 2H), 1.49–1.35 (m, 1H), 1.15–0.99 (m, 2H). ^13^C-NMR (101 MHz, CDCl_3_) δ 165.1, 141.2, 135.9, 134.7, 130.8, 130.4, 128.8, 126.1, 123.0, 59.5, 59.2, 54.3, 46.7, 43.8, 40.7, 35.0, 31.3, 27.8, 26.2, 24.7, 24.5, 23.9, 19.2. ESI-HR-MS: calcd for C_23_H_31_BrN_3_O_3_S [M + H]^+^: 508.1264, found: 508.1249.

*12N-N′-m-Trifluoromethylbenzenesulfonylaminoacetyl aloperine* (**10e**), Yield: 30%; white solid; m.p.: 91–93 °C. ^1^H-NMR (400 MHz, CDCl_3_) δ 8.15 (s, 1H), 8.08 (d, *J* = 7.6 Hz, 1H), 7.84 (d, *J* = 7.6 Hz, 1H), 7.67 (t, *J* = 7.2 Hz, 1H), 5.92 (s, 1H), 5.60 (s, 1H), 4.53 (d, *J* = 4.4 Hz, 1H), 3.92–3.82 (m, 1H), 3.80–3.70 (m, 1H), 3.68–3.57 (m, 1H), 3.28–3.10 (m, 1H), 3.05–2.87 (m, 1H), 2.71–2.60 (m, 1H), 2.59–2.50 (m, 1H), 2.28–2.13 (m, 2H), 2.08–2.00 (m, 1H), 1.99–1.88 (m, 2H), 1.88–1.75 (m, 3H), 1.75–1.66 (m, 2H), 1.65–1.56 (m, 3H), 1.47–1.37 (m, 1H), 1.13–0.99 (m, 2H).^13^C-NMR (101 MHz, CDCl_3_) δ 165.0, 140.8, 134.6, 131.8, 130.8, 130.1, 129.5, 128.8, 124.5, 123.3, 59.5, 59.2, 54.3, 46.7, 43.9, 40.7, 34.9, 31.3, 27.7, 26.2, 24.7, 24.4, 23.9, 19.1.ESI-HR-MS: calcd for C_24_H_31_F_3_N_3_O_3_S [M + H]^+^: 498.2033, found: 498.2013.

*12N-N′-m-Trifluoromethoxybenzenesulfonylaminoacetyl aloperine* (**10f**), Yield: 26%; white solid; m.p.: 81–83 °C. ^1^H-NMR (400 MHz, CDCl_3_) δ 7.83 (d, *J* = 7.2 Hz, 1H), 7.73 (s, 1H), 7.61–7.53 (m, 1H), 7.43 (d, *J* = 8.0 Hz, 1H), 5.93–5.76 (m, 1H), 5.61 (s, 1H), 4.54 (d, *J* = 4.4 Hz, 1H), 3.93–3.81 (m, 1H), 3.79–3.69 (m, 1H), 3.69–3.58 (m, 1H), 3.30–3.11 (m, 1H), 3.03–2.90 (m, 1H), 2.79–2.61 (m, 2H), 2.59–2.50 (m, 1H), 2.28–2.15 (m, 2H), 2.12–2.04 (m, 1H), 2.01–1.90 (m, 2H), 1.89–1.76 (m, 3H), 1.73–1.57 (m, 4H), 1.48–1.35 (m, 1H), 1.14–0.98 (m, 2H). ^13^C-NMR (101 MHz, CDCl_3_) δ 165.1, 149.4, 141.6, 134.7, 131.0, 128.8, 125.7, 125.1, 120.4, 120.1, 59.5, 59.1, 54.4, 46.7, 43.9, 40.7, 34.8, 31.3, 27.7, 26.2, 24.7, 24.4, 23.9, 19.1.ESI-HR-MS: calcd for C_24_H_31_F_3_N_3_O_4_S [M + H]^+^: 514.1982, found: 514.1967.

*12N-N′-(3,5-**Ditrifluoromethyl)benzenesulfonyl**aminoacetyl aloperine* (**10g**). Yield: 22%; brown solid; m.p.: 78–80 °C. ^1^H-NMR (400 MHz, CDCl_3_) δ 8.33 (s, 2H), 8.06 (s, 1H), 6.05 (s, 1H), 5.61 (d, *J* = 4.8 Hz, 1H), 4.55 (d, *J* = 5.2 Hz, 1H), 4.01–3.86 (m, 1H), 3.83–3.75 (m, 1H), 3.73–3.60 (m, 1H), 3.22–3.10 (m, 1H), 3.04–2.91 (m, 1H), 2.76–2.61 (m, 2H), 2.59–2.47 (m, 1H), 2.27–2.17 (m, 2H), 2.07–2.01 (m, 1H), 2.00–1.90 (m, 2H), 1.89–1.77 (m, 3H), 1.76–1.68 (m, 2H), 1.64 (s, 1H), 1.60 (s, 1H), 1.44–1.39 (m, 1H), 1.14–0.99 (m, 2H). ^13^C-NMR (101 MHz, CDCl_3_) δ 164.8, 142.7, 134.6, 132.9 (2), 127.7 (2), 126.3 (2), 122.6 (2), 59.6, 59.2, 54.3, 46.6, 43.9, 40.8, 34.9, 31.2, 27.7, 26.2, 24.7, 24.4, 23.9, 19.1. ESI-HR-MS: calcd for C_25_H_30_F_6_N_3_O_3_S [M + H]^+^: 566.1907, found: 566.1897.

### 3.3. Biology Assay

#### 3.3.1. Cell Culture and Screening of Compounds

Cells were maintained on Dulbecco’s Modified Eagle’s medium (DMEM), laid on 96-well plate. DMEM medium was supplemented with 10% fetal bovine serum (FBS) at 37 °C humidified atmosphere with 5% CO_2_. Furthermore, serum-free culture was required until the cells were confluent at 90−95%. 24 h later the cells were treated with a tested compound (40 and 80 μM) for 24 h. The COL1A1 promotor activity was detected by the Bright-Glo luciferase assay system. Alternatively, to test the acute effects of compound on luciferase activity, 24 h later the cells were lysed and the lysate was treated with a tested compound (80 μM) for 2 h, then the luciferase was detected by the Bright-Glo luciferase assay system (Promega, Madison, WI, USA).

#### 3.3.2. Cell Survival Assay

HepG2 cell survival was measured by MTT assay. HepG2 cells, at density of 6x10^3^ cells/well in 96-well culture plates, were incubated with the aloperine derivates at various concentrations until the cells at 50% confluence for 24 h. 20 µL of the MTT (5 mg/mL) solution was added to each plate and incubated with 5% CO_2_, 37 °C for 4 h. Subsequently, withdrawing the culture supernatant containing MTT, 150 µL DMSO was used for dissolving the formazan crystal. The absorbance was measured using a microplate reader at 570 nm.

LX-2 cell survival was measured by sulforhodamine B (SRB) assay. The cells were seeded in 96-well plates at the density of 2 × 10^4^/well and cultured in DMEM/GlutaMAX I medium with 10% fetal bovine serum (FBS) with 5% CO_2_, 37 °C. The cells were treated by the aloperine derivates for 24 h, washed by PBS for three times. Then the cells were fixed with 10% (*wt*/*vol*) trichloroacetic acid for 1 h and stained with SRB for 0.5 h. Subsequently, the protein-bound dye was dissolved in 10 mM Tris base solution after removing excess SRB dye washing repeatedly with 1% (*vol*/*vol*) acetic acid, the optical density (OD) was determined at 510 nm with a microplate reader.

#### 3.3.3. RT-qPCR Assay

LX-2 cells were laid on 6-well plate and cultured in DMEM supplemented with 10% fetal bovine serum (FBS) in a 5% CO_2_ atmosphere at 37 °C. And serum-free culture was required until the cells reached 90−95% confluence. After 24 h, cells were incubated with 40 and 80 μM concentrations of aloperine derivatives in the presence of 2 ng/mL of TGF-β1 for 24 h. Total RNA of the LX-2 cells was isolated using TRIzol reagent, purified using NucleoSpin RNA Clean-up. Reverse transcription was performed with Transcriptor first strand cDNA synthesis kit. Subsequently, the cDNA levels was determined by ABI 7500 Fast Real-Time PCR System using TaqMan probes of COL1A1, TGF-β1, α-SMA, and GAPDH (sequence not disclosed by ABI) and FastStart Universal Probe master mix (Roche, Indianapolis, IN, USA).

#### 3.3.4. Western Blot

LX-2 cells were cultured as described above. Cells were washed with phosphate-buffered saline (PBS) and were resuspended by radioimmunoprecipitation assay (RIPA) lysis buffer for 30 min in 4 °C; the supernatant was collected after centrifugation at 12000 g for 15 min at 4 °C. Subsequently, equal amounts of protein were quantified with Bradford assay, separated using SDS-PAGE and transferred onto polyvinylidene difluoride membrane. The membranes were blocked with 5% milk in PBST for 1 h, and probed with specific primary antibodies overnight at room temperature. The membrane was washed three times by PBST, and probed with horseradish peroxidase-conjugated secondary antibodies and GAPDH. The proteins were visualized by chemiluminescence reagents. Antibodies used in western blot analysis were obtained from Abcam (Cambridge, UK): anti-collagen 1 (ab34710), anti-TGF beta 1 antibody (ab179695), anti-alpha smooth muscle actin (ab32575) and Cell Signaling Technology (Danvers, MA, USA): GAPDH (D16H11) Rabbit mAb (5174).

#### 3.3.5. Statistics

Results are presented as mean values ± standard error of independent triplicate experiments. All statistical analyses were performed by using two-tailed Student’s t-test and *p*-values of less than 0.05 were considered statistically significant.

### 3.4. Acute Toxicity

Male and Female Kunming mice with weight of 20.0 ±1.0 g were obtained from the Institute of Laboratory Animal Science (Beijing, China). Animals wer e cared according to the institutional guidelines of the Institute of Medicinal Biotechnology, CAMS&PUMC (IMB-20190610D3), and fed with regular rodent chow and housed in an air conditioned room. The mice were randomly divided into different groups with four male mice and four female mice each. Each compound was given orally in a single-dosing experiment, the dose of **4i** was 100, 300 and 400 mg kg^−1^, the dose of **4n** was 300, 400, 500 mg kg^−1^, the dose of **4p** was 200, 400, 500 mg kg^−1^ (ddH_2_O as control). The mice were closely monitored for 7 days. Body weight as well as survival was monitored.

## 4. Conclusions

To conclude, twenty seven new aloperine derivatives with a unique endocyclic scaffold were designed, synthesized and evaluated for their inhibitory effect on COL1A1 promotor in LX2 cells, taking **1** as the lead. The SAR results indicated that the introduction of suitable substituents on the 12*N* atom might be beneficial for activity. Compound **4p** exhibited good potency on COL1A1 promotor with an IC_50_ value of 16.5 μM. Its inhibitory activity against COL1A1 was further confirmed at both the mRNA and protein levels. Meanwhile, it effectively inhibited the expression of fibrogenic proteins, such as α-SMA and TGF-β1, indicating the promise against liver fibrogenesis. It also exhibited a good in vivo safety profile, with an LD_50_ value of 400 mg kg^−1^ in mice via intragastrical administration. Overall, this study initiated the anti-COL1A1 SAR of aloperine derivatives, and thus gave useful information for further developing these compounds into promising anti-liver fibrogenesis candidates with a unique endocyclic scaffold, and compound **4p** has been chosen as the lead for the further research.

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
