# Peer review of "Structure–Activity Relationship of Aloperine Derivatives as New Anti–Liver Fibrogenic Agents"

_molecules, 2020, doi:10.3390/molecules25214977_

Round 1
Reviewer 1 Report
The manuscript is somewhat improved. I have nevertheless the following points:
1. I still do not see how you calculated IC50 in Fig. S2 compound 4h.
2. The effect of compounds on Luciferase are quite close (factor 2-3) to those on cytotoxicity. This renders the utility of the compounds highly questionable. I have that this is sufficient to treat individuals for fibrosis without killing them. Table two lacks statistical numbers.
3. Moreover, there was no effect of the compounds on the reporter after 2 h of incubation. How do you think the compounds act on the cells?
4. The experiments in the in vivo fibrosis models have not been performed. Response: Thanks for the advice. The in vivo anti-fibrosis effects of key compounds in animal models will be done and presented in our future paper.
5. Introduction: How should compounds down-regulate the expression of a promoter? To my opinion promoters are regulatory sequences in the DNA that are not expressed.
Author Response
- The manuscript is somewhat improved. I have nevertheless the following points: I still do not see how you calculated IC50 in Fig. S2 compound 4h.
Response: Thanks for the question. We have performed the IC50 calculation in triplicate experiment for two times, and the same results were obtained as shown in Fig. S2. The dose-dependent linearity was not gained, which was probably due to the its high cytotoxicity (Fig. S3). Therefore, IC50 value of compound 4h was deleted, and the explanation for this phenomenon was supplemented in the revised version, too.
- The effect of compounds on Luciferase are quite close (factor 2-3) to those on cytotoxicity. This renders the utility of the compounds highly questionable. I have that this is sufficient to treat individuals for fibrosis without killing Table two lacks statistical numbers.
Response: Thanks for the question. The key compound 4p gave an IC50 value and CC50 value of 16.5 μM,and 43.7 μM, respectively. As shown in Figure S3, compound 4p gave the cell survival rates of 96.6 and 90.0% at the concentration of 10 and 20 μM, respectively, therefore, compound 4p rarely exerts cell-killing effect at its IC50 concentration of 16.5 μM. The statistical numbers were supplemented in the revised version.
- Moreover, there was no effect of the compounds on the reporter after 2 h of incubation. How do you think the compounds act on the cells?
Response: Thanks for the question. The negative acute effects of key compounds on luciferase activity indicated their direct effects on COL1A1 promotor. And we have added this comment in the revised version.
- The experiments in the in vivo fibrosis models have not been performed.
Response: Thanks for the advice. The in vivo anti-fibrosis effects of key compounds in animal models will be done and presented in our future paper.
- Introduction: How should compounds down-regulate the expression of a promoter? To my opinion promoters are regulatory sequences in the DNA that are not expressed.
Response: Thanks for the criticism. Yes, the original statement is not correct, and we have corrected it into “down-regulate the transcription of COL1A1 gene”, “down-regulate COL1A1 expression” or “inhibition upon COL1A1 expression”, and these changes were highlighted in red in the revised version.
Reviewer 2 Report
Most of my previous concerns have been addressed.
However, there seem to be a few mis-citations, for example references 11 and 12 do not show evidence of Collagen I being a specific marker for Myofibroblasts. Btw, Collagen I is not a specific marker for Myofibroblasts, so this part of the Introduction section is not correct.
I cannot find text relating the findings to the previous publications in the research field, or much critical appraisal.
I cannot understand the statistical analysis. e.g. it is not clear what the “n” is in the statistical analysis is exactly…the information provided is “independent triplicate experiments.” But it is not clear what the authors mean by this, do they mean three independent repeats, or do they mean triplicates, or both? Nor is it clear how many mice were used in the analysis. Unless this is clarified, it is not possible to properly assess the data.
The M&M section does not provide any information on how the Western blot data was quantified (name of software/equipment, etc).
I suggest the authors address these concerns.
Author Response
- Most of my previous concerns have been addressed. However, there seem to be a few mis-citations, for example references 11 and 12 do not show evidence of Collagen I being a specific marker for Myofibroblasts. Btw, Collagen I is not a specific marker for Myofibroblasts, so this part of the Introduction section is not correct.
Response: Thanks for the criticism. We have corrected the mistakes in both citation and expression in the revised version.
- I cannot find text relating the findings to the previous publications in the research field, or much critical appraisal.
Response: Thanks for the kind reminder. Yes, it is a pity that our research has not attracted more attention from our peers. However, our model is now available for external screening service of anti-fibrosis drugs, and we are on the way to obtain our space in this filed.
- I cannot understand the statistical analysis. e.g. it is not clear what the “n” is in the statistical analysis is exactly…the information provided is “independent triplicate experiments.” But it is not clear what the authors mean by this, do they mean three independent repeats, or do they mean triplicates, or both? Nor is it clear how many mice were used in the analysis. Unless this is clarified, it is not possible to properly assess the data.
Response: I am sorry for the confusions. All experiments in the manuscript were preformed for three times, so “independent triplicate experiments” is not correct and we have revised it accordingly. In the acute toxicity experiment, we have clarified that “The mice were randomly divided into different groups with 4 male mice and 4 female mice each”.
- The M&M section does not provide any information on how the Western blot data was quantified (name of software/equipment, etc).
Response: We have added “The gel images were analyzed using the Gel Image System Version 4.2 from Bio-tanon” in the revised version.
This manuscript is a resubmission of an earlier submission. The following is a list of the peer review reports and author responses from that submission.
Round 1
Reviewer 1 Report
Anti-fibrotic therapy is a major unfulfilled demand. In the manuscript by Kun et al., Aloperine derivatives were produced and examined for their anti-fibrotic effects in a stellate cell line. Some compounds were found with increased anti-fibrotic potential. The study is somewhat superficial and does not allow clear conclusions. I have the following comments:
- It is unclear how the substances were dissolved and applied to the cells. Dissolved in DMSO? In which volume per ml of cell culture medium? Did you include the same amount of solvent in all controls?
- It is unclear how the IC50 for inhibition of promotor activity were determined. Figures are not shown. The same holds for cytotoxicity measurements.
- Luciferase assay: what did you measure to control for unspecific inhibition?
- Did the compounds exert cytotoxic effects due to inhibition of collagen promotor activity? Is cytotoxicity desirable? How do you exclude that the effect of your compounds are not simply cytotoxicity? Toxicity of the compounds should be performed in other cell lines to shed light on this point.
- Related: is inhibition of the collagen promotor expected to influence the cell viability?
- Was the effect of the compounds on promotor activity an acute effect? I would expect that if the effect of the compounds was directly on the promotor activity.
- Figure 2: it is unclear what the numbers are.
- Could you comment more on the stimulatory effect of TGFbeta on TGFbeta mRNA? Is this already known?
- Figure 3: the number of independent experiments is missing. Also, it is unclear if 4n and 4p had any effects on TGFbeta and alphaSMA as the loading control was overexposed. I do not see clear effects here.
- The number of independent experiments is not stated anywhere. Statistical significance is not given.
- After sufficient characterization of the compounds in vitro, compounds could be tested in a fibrosis animal model.
Author Response
Point-to-point Responses to Reviewer 1
- It is unclear how the substances were dissolved and applied to the cells. Dissolved in DMSO? In which volume per ml of cell culture medium? Did you include the same amount of solvent in all controls?
Response: Thanks for the questions. The preparation method of substance-containing solutions was supplemented in the method and material part in the revised paper. 0.001 mL per ml test solution of cell culture was applied, and the same amount of solvent was applied in all controls and test samples.
- It is unclear how the IC50 for inhibition of promotor activity were determined. Figures are not shown. The same holds for cytotoxicity measurements.
Response: Thank you for the advice. The determination method was added in the part of 3.3.1. Cell Culture, Screening and IC50 Determination in the revised version. And the bar graphs for IC50 and CC50 determinations were uploaded in a separate supplementary document.
- Luciferase assay: what did you measure to control for unspecific inhibition?
Response: Thank you for the advice. The fluorescence values in the background and the blank control were determined in the experiments to control for unspecific inhibition.
- Did the compounds exert cytotoxic effects due to inhibition of collagen promotor activity? Is cytotoxicity desirable? How do you exclude that the effect of your compounds are not simply cytotoxicity? Toxicity of the compounds should be performed in other cell lines to shed light on this point.
Response: Thank you very much for the constructive questions. It is assumed that the cytotoxic effects of the test compounds might also come from their structure since aloperine has exerted antitumor effects in literatures. Therefore, the inhibition of collagen promotor activity might have little impact on the cell viability. Our compounds are not simply cytotoxic in this paper, because they might have impacted downregulate the transcription level of COL1A1 through an unknown mechanism, which is consistent with its anti- pulmonary fibrosis activity reported, as referred to in this paper. LX2 was applied as the cell model in the anti-COL1A1 evaluation, in practice, after being treated by the tested compounds for 24 hours, the survival states of the cells were observed by microscope before the luciferase detection, and the similar cytotoxic trends were observed (data not shown).
- Related: is inhibition of the collagen promotor expected to influence the cell viability?
Response: We have performed the anti-COL1A1 experiments in multiple of structural scaffolds as disclosed in ref [14-16] in our previous studies, and not all compounds with high anti-COL1A1 activity showed high cytotoxicity. Therefore, inhibition of the collagen promotor seemed to have little impact on cell viability, and the different scaffold might matter.
- Was the effect of the compounds on promotor activity an acute effect? I would expect that if the effect of the compounds was directly on the promotor activity.
Response: It is a good question, and we are on our way to investigate whether it is direct and acute or not.
- Figure 2: it is unclear what the numbers are.
Response: Thanks for the suggestion. We have added the numbers in Figure 2 in the revised version, as introduced in the context.
- Could you comment more on the stimulatory effect of TGFbeta on TGFbeta mRNA? Is this already known?
Response: Thanks for the pertinent comments. Multiple studies have disclosed the stimulatory effect of TGFbeta on TGFbeta mRNA during fibroblast wound healing [1,2], and related comments and references have been added in the revised version.
[1] Song Q. H.; Klepeis V. E.; Nugent M. A.; Trinkaus-Randall V. TGF-beta1 regulates TGF-beta1 and FGF-2 mRNA expression during fibroblast wound healing. Mol Pathol. 2002, 55, 164-76.
[2] Krizhanovsky V, Yon M, Dickins RA, Hearn S, Simon J, Miething C, Yee H, Zender L, Lowe SW. Senescence of activated stellate cells limits liver fibrosis. Cell. 2008 Aug 22;134(4):657-67.
- Figure 3: the number of independent experiments is missing. Also, it is unclear if 4n and 4p had any effects on TGFbeta and alphaSMA as the loading control was overexposed. I do not see clear effects here.
Response: Thanks for the good question. We are very sorry for the mistakes, and we have conducted independent triplicate experiments. New data was added in the revised version.
- The number of independent experiments is not stated anywhere. Statistical significance is not given.
Response: Thanks for the suggestion, and we have corrected the mistakes in the revised version.
- After sufficient characterization of the compounds in vitro, compounds could be tested in a fibrosis animal model.
Response: Thanks for your very constructive advice. We are planning to test the anti-fibrosis effects of 4n and 4p in the rat model of liver fibrosis induced by bile duct ligation, and the results will be presented in our future papers.

Reviewer 2 Report
In this manuscript, Wang kun '[sic]' et al synthesized 27 new aloperine derivatives, and investigated their effect on collagen α1 (COL1A1) promotor activity, mRNA and protein levels, as well as their acute toxicity in vivo. Thereby, they identified two compounds that inhibited collagen α1, showing good in vivo safety, that can be used for future development of drugs that target liver fibrinogenesis. Although the findings are interesting, and useful to develop future research, the Manuscript would need to be significantly modified before it can be published.
Major points:
- The introduction would benefit from a more balanced, clear and correct overview of the research field. For example, the role of α-SMA in the phrase below should be clarified. “Type I collagen (COL1), together with and α-smooth muscle actin(α-SMA) '[sic]' , expressed by myofibroblasts, constitutes the main source of of extracellular matrix (ECM) in all clinical and experimental liver fibrosis [11,12].”
- I cannot find discussion sections relating the findings to the previous publications in the research field, or any critical appraisal. This would need to be added.
- The citations appear to be too much dependent upon self-citations, and I would suggest the authors cite many other researchers in the field of liver fibrosis.
- Parts of the section “2. Results and Discussion” describe details on methods or materials would preferably be moved to the Method and Material section. Example: “Aloperine was purchased from the Yanchi Dushun Biological and Chemical Co. Ltd. (Shanxi, 60 China), with the purity over 95%.
- The heading of the Method and Material section is “Experimental Section”, which is slightly confusing, since experiments were performed also in the previous section in the manuscript. I suggest the authors to change the title to “Method and Material Section”.
- I cannot see the required statistical analysis, for example for the in vivo and PCR results, and suggest the authors add a section describing this in the M&M section.
- Information about the number of biological or technical would be needed to be added. In particular, repeats of the Western blot and PCR experiments, and preferably with quantifications of the Western blot data.
- In line with this, the in vivo data graph should show p-values.
- Also, Figure 2 should show error bars and p-values of the differences observed.
- Because of the many signs of not a fully unbiased analysis and approach, I have concerns regarding the scientific stringency and soundness of the study. For example text like this: “Unfortunately, none of them showed exciting improvement on activity”. Hence, I am not convinced the study was performed in an unbiased, objective and scientific manner.
- The language is very far from being written in Scientific English. For example, “….induced the boomed expression of COL1A1.” A professional scientific English editor would need to edit/re-write the manuscript in Scientific English.
- In addition, there are many typo, formatting, grammatical and spelling errors that the editor should correct. For example, the excessive use of bold font, numbers used as names; example “…..1 was taken”, and “The anti-COL1A1 analysis also witnessed an overall elevation in activity”.
Author Response
Point-to-point Responses to Reviewer 2
1.The introduction would benefit from a more balanced, clear and correct overview of the research field. For example, the role of α-SMA in the phrase below should be clarified.“Type I collagen (COL1), together with and α-smooth muscle actin(α-SMA) '[sic]' , expressed by myofibroblasts, constitutes the main source of of extracellular matrix (ECM) in all clinical and experimental liver fibrosis [11,12].”
I cannot find discussion sections relating the findings to the previous publications in the research field, or any critical appraisal. This would need to be added.
The citations appear to be too much dependent upon self-citations, and I would suggest the authors cite many other researchers in the field of liver fibrosis.
Response: Thank you very much for the constructive advice. We have supplemented the role of α-SMA in the introduction and result and discussion part. We add a new 2.7 Discussion to explain the anti-fibrosis study from other groups as well as connection with our study in the revised version.
- Parts of the section“2. Results and Discussion”describe details on methods or materials would preferably be moved to the Method and Material section. Example:“Aloperine was purchased from the Yanchi Dushun Biological and Chemical Co. Ltd. (Shanxi, 60 China), with the purity over 95%.
Response: Thanks for the good suggestion. We have made the corrections accordingly in the revised version.
- The heading of the Method and Material section is “Experimental Section”, which is slightly confusing, since experiments were performed also in the previous section in the manuscript. I suggest the authors to change the title to “Method and Material Section”.
Response: Thanks for the good suggestion. We have made the corrections accordingly in the revised version.
- I cannot see the required statistical analysis, for example for the in vitro and PCR results, and suggest the authors add a section describing this in the M&M section.
Response: Thanks for the question. It was our mistake to miss the statistical analysis. We have added the section 3.3.5. Statistics in the revised version.
- Information about the number of biological or technical would be needed to be added. In particular, repeats of the Western blot and PCR experiments, and preferably with quantifications of the Western blot data.
Response: Thanks for the pertinent comments. We have supplemented the data in the revised version.
- In line with this, the in vitro data graph should show p-values.
Response: Thanks for the suggestion. We have supplemented the p-values accordingly in the revised version.
- Also, Figure 2 should show error bars and p-values of the differences observed.
Response: Thanks for the suggestion. We have added the error bars and p-values in Figure 2 in the revised version.
- Because of the many signs of not a fully unbiased analysis and approach, I have concerns regarding the scientific stringency and soundness of the study. For example text like this:“Unfortunately, none of them showed exciting improvement on activity”.Hence, I am not convinced the study was performed in an unbiased, objective and scientific manner.
Response: Thank you very much for the constructive advice. We are very sorry for this, and we made revisions accordingly. We have changed the sentence “Unfortunately, none of them showed exciting improvement on activity” into “compounds 7a–d showed a slight increase in activity”, and et al.
- The language is very far from being written in Scientific English. For example,“….induced the boomed expression of COL1A1.”A professional scientific English editor would need to edit/re-write the manuscript in Scientific English.
Response: Thank you very much for the constructive advice. We are very sorry for this, and we have made revisions accordingly.
- In addition, there are many typo, formatting, grammatical and spelling errors that the editor should correct. For example, the excessive use of bold font, numbers used as names; example“…..1 was taken”,and“The anti-COL1A1 analysis also witnessed an overall elevation in activity”.
Response: Thank you very much for the constructive advice. We are very sorry for the mistakes. Based on your suggestion, we have checked the manuscript and revised the mistakes accordingly.
Round 2
Reviewer 1 Report
The manuscript is somewhat improved now and partially reacted to my concerns. However, additional experimentation has not been done, but is certainly be required to result in reliable data.
1. IC50 luciferase activity: many of the curves are not suitable to calculate an IC50. These experiments need to be repeated. The IC50 determination in Fig. S2 are also questionable.
2. Cytotoxicity of the compounds need to be performed in LX-2 cells to conclude that the suggested anti-fibrotic effect is not due to cytotoxicity in the collagen-producing cells.
4. I also suggest to include the answer the question experimentally whether the compounds have acute effects on luciferase activity.
5. The experiments in the in vivo fibrosis models have not been performed.
6. How did the authors come to the idea that the compounds might act through inhibition of transcription? What is the evidence that the compound does not act through inhibition of TGFbeta signalling rather than col2 promotor acitivity?
Line 175: incomplete sentence.
Reviewer 2 Report
The manuscripts still has a few major issues;
- The language is very far from being acceptable Scientific English, and I strongly advice the authors to contact a professional Scientific English language editor (a native english speaker would not be sufficient).
- The statistical information in the figures and figure legends are not clear to me, why are some values indicated with "#" and some with "*"?, I suggest the authors modify this so it is in line with standard formats (thus only use "*").
- I can almost not read the text in the figures, because the font is too small. I therefore suggest that the authors check that the figures are in line with the required format.
- I suggest the authors show all obtained p-values, for example in a separate Supplementary table. Also, they would need to indicate exactly what values they compared, thus, what they used as "N" in their statistical analysis. Preferentially, in the figure legends.